# Proteomic Characterization of a Lunasin-Enriched Soybean Extract Potentially Useful in the Treatment of *Helicobacter pylori* Infection

**DOI:** 10.3390/nu16132056

**Published:** 2024-06-27

**Authors:** Giselle Franca-Oliveira, Sara Monreal Peinado, Stephanny Miranda Alves de Souza, Dario Eluan Kalume, Theo Luiz Ferraz de Souza, Blanca Hernández-Ledesma, Adolfo J. Martinez-Rodriguez

**Affiliations:** 1Institute of Food Science Research (CIAL, CSIC-UAM, CEI UAM + CSIC), Nicolás Cabrera 9, 28049 Madrid, Spain; g.f.oliveira@csic.es (G.F.-O.);; 2Faculdade de Farmácia, Universidade Federal do Rio de Janeiro, Rio de Janeiro 21941-902, Brazil; stephannymas@gmail.com (S.M.A.d.S.); theo.ff.ufrj@gmail.com (T.L.F.d.S.); 3Programa de Pós-Graduação em Nanobiossistemas, Universidade Federal do Rio de Janeiro, Duque de Caxias 25240-005, Brazil; 4Laboratório Interdisciplinar de Pesquisas Médicas, Instituto Oswaldo Cruz, Fundação Oswaldo Cruz, Rio de Janeiro 21041-250, Brazil

**Keywords:** *Helicobacter pylori*, proteomics, soybean, lunasin, oxidative stress, inflammation

## Abstract

*Helicobacter pylori* infection affects over 50% of the world’s population and leads to chronic inflammation and gastric disorders, being the main pathogen correlated to gastric cancer development. Increasing antibiotic resistance levels are a major global concern and alternative treatments are needed. Soybean peptides and other compounds might be an alternative in the treatment to avoid, eradicate and/or control symptoms of *H. pylori* infection. This study aimed to characterize a lunasin-enriched soybean extract (LSE) using proteomics tools and to evaluate its antioxidant, anti-inflammatory and antibacterial properties against *H. pylori* infection. By LC–MS/MS analysis, 124 proteins were identified, with 2S albumin (lunasin and large-chain subunits) being the fourth most abundant protein (8.9%). Lunasin consists of 44 amino acid residues and an intramolecular disulfide bond. LSE at a low dose (0.0625 mg/mL) reduced ROS production in both *H. pylori*-infected and non-infected AGS gastric cells. This led to a significant reduction of 6.71% in the levels of pro-inflammatory interleukin (IL)-8. LSE also showed antibacterial activity against *H. pylori*, which can be attributed to other soybean proteins and phenolic compounds. Our findings suggest that LSE might be a promising alternative in the management of *H. pylori* infection and its associated symptoms.

## 1. Introduction

*Helicobacter pylori* (*H. pylori*) is a gram-negative, microaerophilic, flagellated pathogen that infects the gastric epithelial lining, generating a host’s immune response. It is estimated that more than 50% of the world population is infected by *H. pylori*, although the global prevalence varies according to regions and demographic groups worldwide [1,2]. Although the infection with this pathogen can happen throughout the whole lifespan, it generally occurs in the early stages of human life and will not disappear unless diagnosed and treated [3]. The severity of the infection is conditioned by multiple variables, such as genetic factors, the gastric physiology of the host and the characteristics of the pathogen itself [3,4], and it can be symptomatic or asymptomatic. The clinical manifestations can vary from mild dyspeptic symptoms to severe peptic ulcers and gastric cancer [5,6]. To understand the factors that drive the progression of the disease, the findings in molecular and cellular research of *H. pylori*-induced physiopathological changes are helping to elucidate the mechanisms adopted by this pathogen to manipulate signaling pathways and evade immune responses for the successful colonization of the gastric mucosa [7]. Furthermore, the dynamics between *H. pylori* and the gut microbiota can also influence systemic health, causing problems not only at the gastrointestinal level [8,9].

The immune response raised by *H. pylori* infection induces a chronic inflammatory response that is associated with gastric cancer, being the strongest risk factor for the development of this disease [10]. Classified by the International Agency for Research on Cancer as a class I carcinogen, the pathogen is correlated with 80% of gastric cancer cases [11,12]. For this reason, eradicative treatment has been indicated. The treatment is conducted with traditional antibiotic therapies combined with other medicines to assist and diminish side effects. However, these adverse side effects and the rising antibiotic resistance of pathogens highlight the urgency to search for alternative approaches [13,14,15].

Despite the known importance of changes in diet to lead a healthy life specially in the context of diseases, diet therapy is often neglected in clinical practice [16]. Diet directly influences gastrointestinal health and dietary choices can act favoring *H. pylori* proliferation, inducing clinical manifestations and ultimately gastric cancer [17]. Besides diet itself, food-derived bioactive compounds can be allies against the progression of infectious diseases such as *H. pylori* infection. Thus, current studies are focused on demonstrating the biological properties of food-derived compounds as the basis of their use as ingredients of functional foods or nutraceuticals [18].

Soybean is one of the main sources of bioactive compounds, such as phytochemicals, peptides, oligosaccharides, dietary fibers, isoflavones, saponins, phenolic and phytic acids. They are related to several health benefits like cardiometabolic and bone health, cancer prevention, immunomodulation and antiaging effects. Antibacterial activity against *H. pylori* has been reported in some soybean extracts, both fermented and unfermented, with different compounds being considered responsible for the effects [19,20]. In the case of soybean-derived peptides, although they have been reported to exert antimicrobial potential against gram-positive and gram-negative bacteria, no data on their effects against *H. pylori* are still available [20]. The antimicrobial activity attributed to peptides is dependent on their characteristics and structure, but their mechanism of action mostly relies on destabilization of the bacterial cytoplasmic membrane that leads to lysis of the cell [21]. One of the most studied soybean-derived peptides is lunasin, whose multifunctionality has been associated with its anticarcinogenic, anti-inflammatory, antioxidant and hypocholesterolemic properties [22]. Although no antimicrobial effects have been reported for lunasin, its anti-inflammatory and antioxidant properties demonstrated in cell and animal models [23] suggest the potential of this peptide in the control/management of *H. pylori* infection-associated symptoms. Thus, the main objectives of the present work were to characterize the proteome of a lunasin-enriched soybean extract (LSE) and to evaluate its antimicrobial, antioxidant and anti-inflammatory properties as the basis of its potential against *H. pylori* infection.

## 2. Materials and Methods

### 2.1. Materials

Albumin-enriched soybean extract (SE) was supplied by Reliv International Inc. (Chesterfield, MO, USA). 3-(4,5-dimethylthiazol-2-yl)-2,5-diphenyltetrazol bromide (MTT), dimethyl sulfoxide (DMSO), 2′7′-dichlorofluorescein diacetate (DCFH-DA) and tert-butyl hydroperoxide (t-BOOH) were purchased from Sigma-Aldrich (St. Louis, MO, USA). All other reagents were of analytical grade.

### 2.2. Preparation of Lunasin-Enriched Soybean Extract (LSE)

A total of 500 mg of SE was dissolved in 50 mL of PBS, stirred at 4 °C overnight and centrifuged at 1000× *g* for 5 min twice, collecting the supernatant by decantation and further lyophilization (LSE), and kept at −20 °C until further assays. The total protein concentration was determined by Bradford assay in a flat-bottom, 96-well microplate (Kasvi, São José dos Pinhais, PR, Brazil) using Bio-Rad reagent (Bio-Rad Laboratories, Inc., Hercules, CA, USA) according to the manufacturer’s instructions. Bovine serum albumin (BSA) was used as a standard at concentrations ranging from 50 to 1000 µg/mL. LSE samples were prepared in two different concentrations in deionized water. The assays were performed in triplicate. Absorbance measurements at 595 nm were carried out on a SpectraMax M5 microplate reader (Molecular Devices, LLC., San Jose, CA, USA).

### 2.3. Proteomic Analysis of the Lunasin-Enriched Soybean Extract (LSE)

The strategy employed to perform the proteomic analysis of the LSE is schematized in Figure 1. The sample was prepared by dissolving 4 mg of LSE in 250 µL of 0.1 M NH_4_HCO_3_ solution, pH 8.3, to a total protein concentration of 12 µg/µL.

#### 2.3.1. In-Solution Digestion

The enzymatic hydrolysis of LSE was based on the methodology described by Rech et al. [24]. A volume of 100 µL of LSE (12 µg/µL) was collected for in-solution digestion. RapiGest SF surfactant (Waters Corporation, Milford, MA, USA) was added to a final concentration of 0.1% (p/v) and incubated for 15 min at 80 °C under agitation in a thermo shaker (Nova Instruments, NI 1363 model, Piracicaba, SP, Brazil). Then, dithiothreitol (DTT) was added to a final concentration of 10 mM and incubated for 30 min at 60 °C to reduce cysteine residues. Alkylation of the cysteines was performed by adding iodoacetamide (IAA) to a final concentration of 10 mM and reaction was carried out for 30 min at room temperature in the dark. Trypsin digestion was conducted overnight by adding 2 µg of the enzyme. RapiGest SF was precipitated by reacting the sample with 0.5% trifluoroacetic acid (TFA) at 37 °C for 90 min. Afterwards, the sample was centrifugated at 17,530× *g* for 30 min at 5 °C and the supernatant was collected and dried to a final volume of approximately 100 µL in a vacuum centrifuge (Savant SPD111V SpeedVac Concentrator; Thermo Electron Corp., Milford, MA, USA).

#### 2.3.2. Analysis of the In-Solution, Digested, Lunasin-Enriched Soybean Extract (LSE)by Liquid Chromatography Tandem Mass Spectrometry (LC–MS/MS)

The LSE sample from the in-solution digestion was submitted for qualitative and quantitative analysis. For absolute quantification, alcohol dehydrogenase from *Saccharomyces cerevisiae* (Waters Corp.) was used as standard protein and added into the sample to a final concentration of 200 fmol. The tryptic peptides were desalted online using a Waters Symmetry C18 180 μm 20 mm and 5 μm trap column, and a sample injection volume of 1 µL (or 6 µg) was applied in five replicates. Both quantitative and qualitative analyses were performed by LC using an ACQUITY UPLC M-CLASS HSS T3 C18 column, 100 μm × 100 mm, 1.8 μm (Waters Corp.), with a flow of 0.5 μL/min. The mobile phase A (0.1% (*v*/*v*) formic acid in water) and mobile phase B (0.1% (*v*/*v*) formic acid in acetonitrile) were combined in a step gradient according to the following: 3 to 5% solvent B (0 to 3 min) increased to 40% solvent B (3 to 95 min), 40 to 85% solvent B (95 to 96 min), held at 85% solvent B (96 to 106 min), decreased to 3% solvent B (106 to 107 min) and finally held at the gradient initial conditions until the end of the run (130 min). 

The quantitative analyses were carried out on a Waters Synapt G1 HD/MS High-Definition Mass Spectrometer (Waters Corp.) set at MSE mode, following a label-free protein quantification methodology [25,26] and coupled to the Nano ACQUITY UPLC chromatography system. The ESI voltage was set at 3000 V, the source temperature was 80 °C and the cone voltage was 40 V. The MassLynx data system (Version 4.1; Waters Corp.) was utilized to control the instrument and for data acquisition. The experiments were performed by scanning from a mass-to-charge ratio (*m*/*z*) of 50–2000 using a scan time of 0.8 s. MSE label-free quantification acquisitions were performed by alternating the collision energy between 6 V in low energy and ramped from 12 to 35 V in high energy, using argon as the collision gas at a pressure of 40 psi. 

For the qualitative analysis, the mass spectral data were acquired on the same mass spectrometer as described above in data-dependent analysis (DDA) in positive ion detection mode. The ToF mass analyzer was calibrated with [Glu1]-fibrinopeptide B fragment ions in the range of 50 to 2000 *m*/*z*, and the intact double-charged ion at 785.8426 *m*/*z* was used as a reference (“lock mass”) for accurate correction of the mass measurement. A sample volume of 1 µL (or 6 µg) was injected in three replicates into the same LC–MS/MS system described above. The ESI voltage was set at 4000 V, the source temperature was 80 °C and the cone voltage was 40 V. The MassLynx data system (Version 4.1; Waters Corp.) was also used to control the instrument and for data acquisition. The experiments were performed by scanning from a *m*/*z* of 50 to 2000 using a scan time of 1 s and an interscan time of 0.02 s. The three more intense peptide ion signals were selected to fragmentation and the collision energy was automatically adjusted according to the ion *m*/*z* range.

#### 2.3.3. Bioinformatics Analysis

Data were processed using the Progenesis QI for proteomics version 4.2 software platform (Nonlinear Dynamics; Waters Corp.). Protein and peptide tolerance were set as 100 ppm and 50 ppm, respectively. A false discovery rate was set to 1%. All quantified proteins with an ANOVA *p*-value lower than 0.05 were considered as a positive identification for raw abundance analysis and followed by a quantification using the abundance of the top three non-conflicting peptides of the external standard as a reference to estimate the amount (fmol) of the other injected proteins.

For quantitative analysis, raw data were searched against the *Glycine max* Uniprot non-reviewed protein database, including possible contaminants, such as human keratin proteins and trypsin from porcine pancreas. Trypsin enzyme was utilized, and one missed cleavage site was allowed. Cysteine carbamidomethylation was set as a fixed modification, the peptide N-terminal carbamidomethylation, methionine oxidation, and glutamine to pyroglutamic acid at the N-terminal, and asparagine and glutamine deamidation were set as the variable modifications.

For the protein identification (qualitative analysis), raw mass spectral data were processed on a Protein Lynx Global Server (PLGS) version 3.0 licensed software (Waters Corp.) and the processed data were searched against the Uniprot *Glycine max* protein database using the MASCOT search engine (license version 2.8). SemiTrypsin enzyme was utilized, and one missed cleavage site was allowed. The precursor as well as fragment ion tolerances were 1.0 Da and 0.5 Da, respectively. The false-positive discovery rate (FDR) was set at 1%, the significance threshold value *p* < 0.05 and peptide ion score ≥25. Cysteine carbamidomethylation was set as the fixed modification, and carbamidomethylation at the N-terminal, with deamidation of asparagine and glutamine, conversion of N-terminal glutamine or glutamate to pyroglutamic acid, and methionine oxidation set as variable modifications. Biological process and molecular function information were retrieved from the UniProtKB and Gene Ontology databases.

### 2.4. Analysis of the Intact Proteins (2S Albumin and its Subunits) in the Lunasin-Enriched Soybean Extract (LSE)

#### 2.4.1. Separation of Lunasin-Enriched Soybean Extract (LSE) Fractions by Ultrafiltration

To better characterize whether lunasin was dissociated or associated to the large chain of the 2S albumin in the LSE, ultrafiltration was performed with reduced and non-reduced LSE samples. Firstly, 0.1 g of LSE was added in two microcentrifuge tubes each and dissolved in 1 mL of 0.1 M NH_4_HCO_3_ solution, pH 8.0. To reduce the LSE sample, a final concentration of 10 mM DTT was added. LSE samples were submitted to sonication for 6 min at 30 °C for better solubilization. Then, the LSE sample with 10 mM DTT was incubated for 35 min at 37 °C to complete reduction. Both samples—with and without DTT—were then centrifuged (Centrifuge 5804R; Eppendorf AG, Hamburg, Germany) at 13,000× *g* for 15 min at 4 °C. This procedure was repeated three times until the supernatant was reasonably clear. The supernatants were transferred to Amicon^®^ Ultra 0.5 mL centrifugal filter units (Merck Millipore Ltd., Darmstadt Germany) with a 10 kDa cutoff, followed by centrifugation at 13,000× *g* for 10 min at 4 °C. This step was performed three times. The retentate and filtrated fractions of both reduced and non-reduced samples were collected, partially concentrated in a vacuum centrifuge (Concentrator Plus; Eppendorf AG) and kept at −20 °C.

#### 2.4.2. Analysis of the Intact Proteins by LC–MS

The retentate and filtrated fractions of both reduced and non-reduced samples were diluted to approximately 2 µg/µL with 3% (*v*/*v*) acetonitrile and 0.1% (*v*/*v*) formic acid. The samples were centrifugated at 13,500× *g* for 10 min at room temperature, and then, the supernatant was transferred to injection glass vials. For the analyses of the filtrated and retentate fractions, C18 and C4 reversed-phase capillary columns were used, respectively. The filtrated samples (1 µL) were injected into an ACQUITY UPLC M-Class HSS T3 Column, 100 Å, 1.8 µm, 75 µm × 150 mm (Waters Corp.), while the retentate fractions (1 µL) were injected into a nanoACQUITY BEH300 C4 Column, 300 Å, 1.7 µm, 75 µm × 100 mm (Waters Corp.). In both analyses, the flow rate was set at 0.4 μL/min. The mobile phases A and B were the same as described in the Section 2.3.2. and the linear gradient was the following: 3 to 30% solvent B (0 to 0.1 min), held at 30% solvent B (0.1 to 6.25 min) and increased to 60% solvent B (6.25 to 6.26 min), held at 60% solvent B (6.26 to 12.5 min) and increased to 85% solvent B (12.5 to 15.0 min), held at 85% solvent B (15.0 to 18.75 min) and decreased to 3% solvent B (18.75 to 22.5 min), and finally, held at the gradient initial conditions until the end of the run (30 min). Data were analyzed using the MassLynx 4.0 (Waters Corp.) package. Each scan of the chromatogram (elution time) was manually checked to search for the molecular mass corresponding to the intact 2S albumin and its subunits (lunasin and large chain) in both the retentate and filtrated fractions.

### 2.5. Antioxidant and Anti-Inflammatory Activity of the Lunasin-Enriched Soybean Extract (LSE)

#### 2.5.1. Cell Culture

Human cells derived from a stomach adenocarcinoma (AGS) (American Type Culture Collection, ATCC, Rockville, MD, USA), were grown in modified Dulbecco’s Eagle/F12 Medium (DMEM/F12) (Lonza, Basel, Switzerland) and supplemented with 10% (*v*/*v*) fetal bovine serum (FBS) (Hyclone, GE Healthcare, Logan, UK) and 1% penicillin/streptomycin/amphotericin (*v*/*v*) (Lonza). Cells were grown at 37 °C under constant conditions of humidity, 5% CO_2_ and 95% air.

#### 2.5.2. Effects on Cell Viability

The evaluation of the effect of LSE on cell viability was carried out using the MTT assay, commonly used for the measurement of cellular metabolic activity. Cells were seeded onto 96-well plates (Sarstedt AG & Co., Nümbrecht, Germany) at a density of 1 × 10^5^ cells/well and incubated at 37 °C for 24 h. After removing the culture medium, the samples of LSE (0.125 to 5 mg of protein/mL) were dissolved in DMEM/F12 without FBS and added to the wells, and the plate was incubated for 2 and for 24 h at 37 °C. DMEM without FBS was used as a negative control. After discarding the treatment, MTT solution (0.5 mg/mL) was added, and cells were incubated for 2 h at 37 °C. The formazan crystals were dissolved in DMSO, and the absorbance was measured at 570 nm in the reader Synergy HT (BioTek Instruments Inc., Winooski, VT, USA). The results were expressed as a percentage of the control, which was considered 100%.

#### 2.5.3. Cell Infection by *H. pylori*

AGS cells were seeded in plates according to the assay to be conducted. After the pre-treatment with the samples of LSE, cells were infected with a bacterial inoculum (500 μL/well, 1 × 10^8^ colony-forming units (CFU)/mL) of the *H. pylori* strain Hp59. This strain was provided by the Microbiology Department of the University Hospital La Princesa (Madrid, Spain), having been isolated from a gastric biopsy of a symptomatic patient. It is resistant to metronidazole, one of the most widely used antimicrobials in *H. pylori* therapy, and it has pathogenicity attributes associated with high virulence and the most severe clinical conditions [27]. The strain was dissolved in DMEM/F12 without FBS or antibiotics. The infected cells were incubated at 37 °C in a variable atmosphere incubator (VAIN MACS-VA500; Don Whitley Scientific, Bingley, UK) under microaerophilic conditions (85% N_2_, 10% CO_2_, 5% O_2_). For ROS production, the incubation time was of 3 h. For cytokine production, the incubation time was 24 h.

#### 2.5.4. Effects on Reactive Oxygen Species (ROS) Generation

The effects of LSE on the intracellular ROS levels were determined using DCFH-DA as a fluorescent probe. Cells were seeded (1 × 10^5^ cells/well) onto a 24-well plate (Sarstedt, AG & Co.) and incubated at 37 °C for 24 h. After discarding the medium, cells were pre-treated and incubated for 2 h at 37 °C with 500 μL of LSE dissolved in DMEM/F12 without FBS and then filtered through 0.22 µM filter units. The cells were washed with DPBS and 500 µL of DCFH-DA (20 µM, dissolved in DMEM/F12 without FBS) was added to each well and incubated for 30 min. The assayed conditions were AGS cells challenged with t-BOOH as a pro-oxidative agent and AGS cells infected with the Hp59 strain. For the t-BOOH challenged cells, 500 µL of t-BOOH (2.5 mM, dissolved in DMEM/F12 without FBS) was added to each well and incubated for 3 h at 37 °C, with readings every 90 min. For the cells challenged by the infection, 500 µL of bacterial inoculum (previously described) was added to each well and incubated for 3 h at 37 °C, with readings every 60 min. The fluorescence was measured at the excitation and emission wavelengths of 485 nm and 530 nm, respectively, in a Synergy HT plate reader (BioTek Instruments Inc.). The results were expressed as ROS levels (% compared to the controls, which were considered as 100%). For chemically stimulated cells, a negative (basal conditions) and a positive control (chemically challenged) were used, whereas for *H. pylori*-infected cells, only a positive control was used, because the reading was affected by turbidity after infection. The measurement of ROS was conducted using a chemical pro-oxidative agent and with the pathogen itself (*H. pylori*) for comparison and understanding of the response of AGS cells to different kinds of stimuli on ROS production.

#### 2.5.5. Effects on Cytokine Production

AGS cells were seeded (1 mL/well, cell density 2 × 10^5^) in 48-well plates (Corning Costar Corp., Corning, NY, USA) and infected with bacterial inoculum, as previously described. Before infection, the cells were treated (2 h) with LSE. After the infection, supernatants were collected from the wells, centrifuged (10 min at 12,000 rpm) and then stored at −20 °C. Pre-treated and non-infected cells were used as a negative control and AGS cells infected without a pre-treatment were used as a positive control. The cytokine production (interleukin IL-10 and tumor necrosis factor TNF-α) was measured with the enzyme-linked immunosorbent assay (ELISA) using kits, according to the manufacturer’s instructions (eBioscienceTM; Thermo Fisher Scientific). The cytokine IL-8 secretion was measured using a Diaclone ELISA kit (Besançon, France), following the manufacturer’s instructions. The cytokine of main interest was IL-8, due to its relation with *H. pylori* infection. IL-10 and TNF-α were measured to understand if the samples and the infection with the specific strain used were capable of changing cytokine release. 

### 2.6. Antimicrobial Activity against H. pylori of Lunasin-Enriched Soybean Extract (LSE)

*H. pylori* strain Hp59 was suspended in Brucella Broth (Becton, Dickinson, & Co., Madrid, Spain) with 20% glycerol as a cryoprotective agent and stored in cryovials at −80 °C. The reactivation of the strain was performed by inoculation of 200 µL on Müeller–Hinton agar supplemented with 5% defibrinated sheep’s blood (MHB) (Becton, Dickinson, & Co.) and then incubated in a microaerophilic atmosphere using a variable atmosphere incubator (VAIN) (85% N_2_, 10% CO_2_, 5% O_2_) at 37 °C for 72 h. For the experiments using a liquid medium for bacterial growth, Brucella Broth (BB, Becton, Dickinson, & Co.), supplemented with 10% horse serum (HS) (Biowest, Barcelona, Spain), was used. 

The antimicrobial activity of LSE against *H. pylori* was carried out according to the following: 1 mL of LSE samples (final concentration 1 mg/mL) was dissolved in water and filtered through 0.22 μm filter units, mixed with 4 mL of BB supplemented with 10% HS in a conical flask. Bacterial biomass grown in plates was diluted in 2 mL of BB supplemented with 10% HS, under aseptic conditions, and this bacterial inoculum was added (100 μL, ~1 × 10^8^ CFU/mL). The conical flasks were incubated under agitation (150 rpm) in a microaerophilic atmosphere using a VAIN at 37 °C for 24 h. Positive growth controls were prepared with 1 mL of water in 4 mL of BB supplemented with 10% HS and 100 μL of bacterial inoculum. A negative control was prepared with 5 mL of BB supplemented with 10% HS. After incubation, serial decimal dilutions were prepared in saline solution (0.9% NaCl) and 20 μL of each was seeded on plates of MHB agar and incubated in a microaerophilic atmosphere (VAIN) at 37 °C for 120 h. All samples were analyzed in quadruplicate (n = 4). After 120 h of incubation, the number of CFUs was evaluated and the results were expressed as the logarithm of CFU per mL (log CFU/mL).

### 2.7. Statistical Analysis

The obtained results were analyzed using one-way ANOVA, followed by Tukey’s test or Student’s *t*-test, according to the data set. Statistical significance was considered with a *p* ≤ 0.05. The analyses were conducted using the statistical analysis program GraphPad Prism 8.0 (GraphPad Software, San Diego, CA, USA).

## 3. Results and Discussion

### 3.1. Proteomic Characterization of the Lunasin-Enriched Soybean Extract (LSE)

Qualitative analysis of the proteins in the LSE was performed by LC–MS/MS. A Mascot Server search identified a total of 124 proteins, including different isoforms (Appendix A). Predominantly, these proteins were seed storage proteins belonging to the 11S (glycinins), 7S (β-conglycinins) and 2S (2S albumin and napin-type 2S albumin 1) families. Furthermore, the LSE contained several protease inhibitors (e. g., Kunitz and Bowman–Birk trypsin inhibitors), which could protect lunasin from digestion in the gastrointestinal tract, as previously reported [28]. In addition, oleosins and lipoxygenases were also present in the LSE. By using UniProtKB and Gene Ontology, we found that 112 identified proteins had molecular functions (Figure 2A) and that the predominant ones were nutrient reservoirs (17 proteins), oxidoreductases (16 proteins), metal ion binding proteins (15 proteins) and endopeptidase inhibitors (11 proteins), corresponding to 15%, 14%, 13% and 10% of the total proteins, respectively. These proteins were involved in 20 biological processes (Figure 2B), such as response to stress (15 proteins), proteolysis (11 proteins), response to abscisic acid (11 proteins) and lipid metabolic process (7 proteins).

A quantification analysis of the proteins contained in the LSE was performed using the label-free MSE approach (Appendix A). The 2S albumin (lunasin and large-chain subunits) was the fourth most abundant protein (5939.51 fmol) among those detected in the LSE (Table 1). It corresponded to 8.9% of the total proteins quantified. LSE was constituted mainly by the globulin glycinin and β-conglycin subunits (Table 1), which are considered the major components of soybean seed storage proteins [29]. They corresponded to 31.98% and 20.36% of the total proteins quantified in LSE, respectively. These proteins, which together account for more than half of the total amount of LES proteins, have demonstrated antibacterial activity against gram-positive and gram-negative species [30,31].

### 3.2. Structural Analyses of the Lunasin in the Lunasin-Enriched Soybean Extract (LSE)

In our qualitative analysis, we identified peptides comprising the sequences of both small (with a sequence coverage of 88.6%) and large (with a sequence coverage of 76.6%) chains (Figure 3). Notably, a peptide corresponding to the C-terminal of the lunasin, containing an asparagine residue, was also identified. This result indicated that lunasin present in the LSE had a length of 44 amino acid residues, consistent with findings in other studies that extracted lunasin from soybean [32,33]. The lunasin peptide corresponds to the small chain of the mature form of soybean 2S albumin. The precursor 2S albumin sequence constitutes a signal peptide, a small chain, a propeptide and a large chain (Figure 3) [34]. This protein is processed by proteolytical cleavage, and its mature form constitutes the small (lunasin) and large chains connected by disulfide bonds. In a study by Serber et al., the mature form of the 2S albumin (14 kDa) in soybean extract was identified, emphasizing the importance of including a reducing step in the purification of lunasin from soybean extract to recover this peptide separately [33].

To further evaluate whether lunasin was dissociated from or associated with the large chain of the 2S albumin in the LSE, we conducted a fractionation step through ultrafiltration (10 kDa cut-off) with reduced and non-reduced LSE samples. Since 2S albumin has a 14 kDa molecular weight, while lunasin and the large chain are both <10 kDa, we expected to find lunasin in the filtrated fraction only if it was dissociated from the other chain. MS analyses of the intact proteins were performed with the retentate and filtrated fractions of the reduced and non-reduced LSE samples. The filtrated fraction (<10 kDa) of the reduced and non-reduced sample mass spectra showed the presence of ions corresponding to both lunasin and the large chain of the 2S albumin individually (Figure 4 and Figure 5). These results indicate that lunasin was dissociated from the large chain in the LSE. Both lunasin and the large-chain monoisotopic masses indicated that the cysteine residues in these proteins were forming intramolecular disulfide bonds (Figure 4C and Figure 5C). The intact lunasin mass detected in the non-reduced LSE sample (5137.3 Da) corresponds to the theoretical monoisotopic mass of its form with 44 residues (containing a C-terminal asparagine) and an intramolecular disulfide bond. No peak corresponding to the mass of the 2S albumin in any of the retentate samples was identified.

### 3.3. Antioxidant and Anti-Inflammatory Activity of LSE

After AGS cell treatment with LSE, cell viability decreased at all concentrations tested (0.125 to 5 mg of protein/mL) in a dose-dependent manner. Thus, the evaluation of the potential antioxidant and anti-inflammatory activities was conducted with an LSE concentration of 0.0625 mg protein/mL, consistent with a cell viability higher than 80%. Despite the low concentration used, LSE reduced ROS production by 22.7% in comparison to the negative control in basal cells, indicating an antioxidant effect (Figure 6). However, in t-BOOH-challenged AGS cells, LSE did not show any effect. This could be due to the high stability of t-BOOH in aqueous solution, generating a persistent oxidative effect in the cells that could not be reverted by LSE under the assayed conditions [35]. This result differed from some previous studies conducted with different soybean preparations and derived compounds, under different conditions. Thus, a simulated gastrointestinal hydrolyzate from soybean flour reduced ROS production in H_2_O_2_-challenged AGS cells, attributing the effects to medium and short peptides [36]. Our previous studies have demonstrated the ability of synthetic lunasin to inhibit ROS production in different cell lines. Thus, this peptide, at concentrations ranging from 0.5 to 10 µM, reduced ROS levels from 190% to 122% in t-BOOH-induced HepG2 cells [37]. In Caco-2 cells, the antioxidant effect of this peptide was also demonstrated, reducing ROS levels in a dose-dependent manner, with a reduction of 62.7% at 25 µM [38]. Recently, we demonstrated that a soybean protein isolate reduced ROS generation in lipopolysaccharide (LPS)-challenged macrophage RAW264.7, being the medium- and low-molecular weight peptides mainly responsible for the observed effects [39]. The discrepancies between these studies and our current results could be due to different factors such as the type of cell line used, the concentration of lunasin or antioxidant peptides present in the LSE or the presence of other compounds in our extract that could have antagonized the effect of lunasin and other antioxidant compounds [22,40]. 

In contrast to what was observed in chemically induced AGS cells, the LSE significantly reduced the generation of ROS in Hp59-infected cells by 10.4% (Figure 7A). 

In *H. pylori* infection, ROS production plays a major role, having a paradoxical effect. The overproduction of this biomarker by cells aimed at eliminating the pathogen can damage cells of the host and also foster a propitious environment for bacteria to colonize and develop multidrug resistance [41,42]. Thus, controlling ROS levels is of paramount importance. The results obtained indicated that the LSE might have exerted a protective effect against oxidative stress in *H. pylori*-infected cells that could be attributed to proteins/peptides or other bioactive compounds contained in the soybean extract. The antioxidant effects of lunasin and other soybean peptides have been previously reported in different cell and animal models, but not in *H. pylori*-infected cells [43,44]. Furthermore, other compounds present in the LSE and not determined in this work could have contributed to the reduction of ROS production by the cells. Thus, a reduction of ROS levels in AGS cells infected with *H. pylori* was observed in a study conducted with anthocyanins from black soybean at concentrations ranging from 12.5 to 50 µM [45].

The reduction in ROS production in AGS cells caused by the LSE was consistent with the decrease in IL-8 production, the main pro-inflammatory cytokine associated with *H. pylori* infection and the development of gastric cancer [46]. It provoked a significant reduction of 6.71% in the levels of IL-8 produced by AGS cells (Figure 7B). The anti-inflammatory activity of the peptide lunasin is associated with a suppression of the NF-κB pathway by its reduction of the release of cytokines such as IL-1, IL-6 and IL-8 and also the ability to inhibit integrins [44], another important mechanism in *H. pylori* infection. Although there are no studies conducted in *H. pylori*-infected cells with the protein fraction of soybeans, bioactive compounds from sprouted soybean digest reduced the release of IL-8 in Caco-2 cells by 19.5% [47]. Other compounds such as the anthocyanin fraction from black soybean have also been reported to decrease IL-8 levels by 45.8% in the same infected cell model used in our study [45]. LSE did not provoke significant changes in the release of the cytokines TNF-α and IL-10 in Hp59-infected cells. These results suggest that, together with lunasin, other compounds present in the extract may have been involved in the modulation of the inflammatory response provoked by the LSE and should be studied in future research with the purpose of enhancing the anti-inflammatory capacity of this type of extract.

### 3.4. Antimicrobial Activity of LSE against H. pylori

Figure 8 shows the antibacterial effect of LSE at 1 mg protein/mL against *H. pylori*. The extract caused a small but significant decrease of 0.40 log CFU in *H. pylori* growth. It has been described that peptides with greater antibacterial activity are usually of small size (10–50 amino acids) and highly cationic, with a tendency to adopt amphipathic structures, which increases their affinity for bacterial membranes [48]. The lunasin peptide consists of a sequence of 43–44 amino acids, with many hydrophilic and charged residues, and a theoretical pl of 4.43 [49,50]. To date, the antibacterial activity of this peptide has not been reported because of its length and structural properties which do not seem consistent with previously identified antibacterial peptides. However, other proteins or peptides contained in the LSE could have been responsible for the observed effects against *H. pylori*. Thus, glycinins and β-conglycinins, predominant proteins in the LSE (Appendix A) could have contributed to this activity since hydrolysates from these proteins have been demonstrated to exert potent antimicrobial effects against *E. coli* [51]. The Kunitz-type trypsin and Bowman-Birk serine protease inhibitors, also identified in the LSE, have been associated with antibacterial activity [52]. Other soybean-derived peptides such as PGTAVFK and IKAFKEATKKVDKVVVVLWTA have also shown antibacterial activity against other gram-negative bacteria such as *Pseudomonas aeruginosa* [53]. In addition, the phenolic fraction could also contribute since phenolic compounds have been related to the inhibition of *H. pylori* growth exerted by fermented soybean sprouts [19].

## 4. Conclusions

The proteome of LSE revealed the presence of 124 proteins with 2S albumin (lunasin and large-chain subunits) being the fourth most abundant protein. Lunasin in LSE consisted of 44 residues (containing a C-terminal asparagine) and an intramolecular disulfide bond. LSE at 0.0625 mg/mL had an antioxidant effect by reducing ROS production in both *H. pylori*-infected and uninfected AGS gastric cells. Moreover, the production of the pro-inflammatory cytokine IL-8 was significantly reduced after the pre-treatment of infected cells by LSE. The extract also showed antimicrobial activity at 1 mg/mL. The bioactive properties of LSE could be attributed to several soybean proteins contained in the extract, although the contribution of other compounds such as phenolics could not be discarded. Our research suggests that LSE may potentially be a useful tool in the treatment of *H. pylori* gastric infection, diminishing the oxidative stress and inflammation associated with this disease. Further studies are required to explore the role of other components of the extract in the observed behavior.

## Figures and Tables

**Figure 1 nutrients-16-02056-f001:**
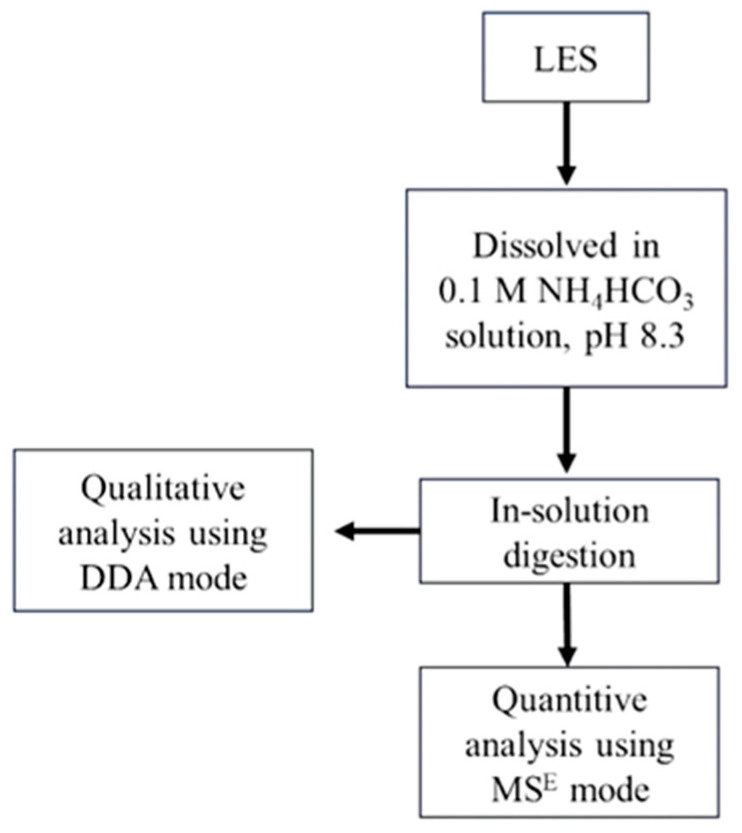
Schematic representation of the methodology employed for lunasin-enriched soybean extract (LSE) characterization using proteomic analysis.

**Figure 2 nutrients-16-02056-f002:**
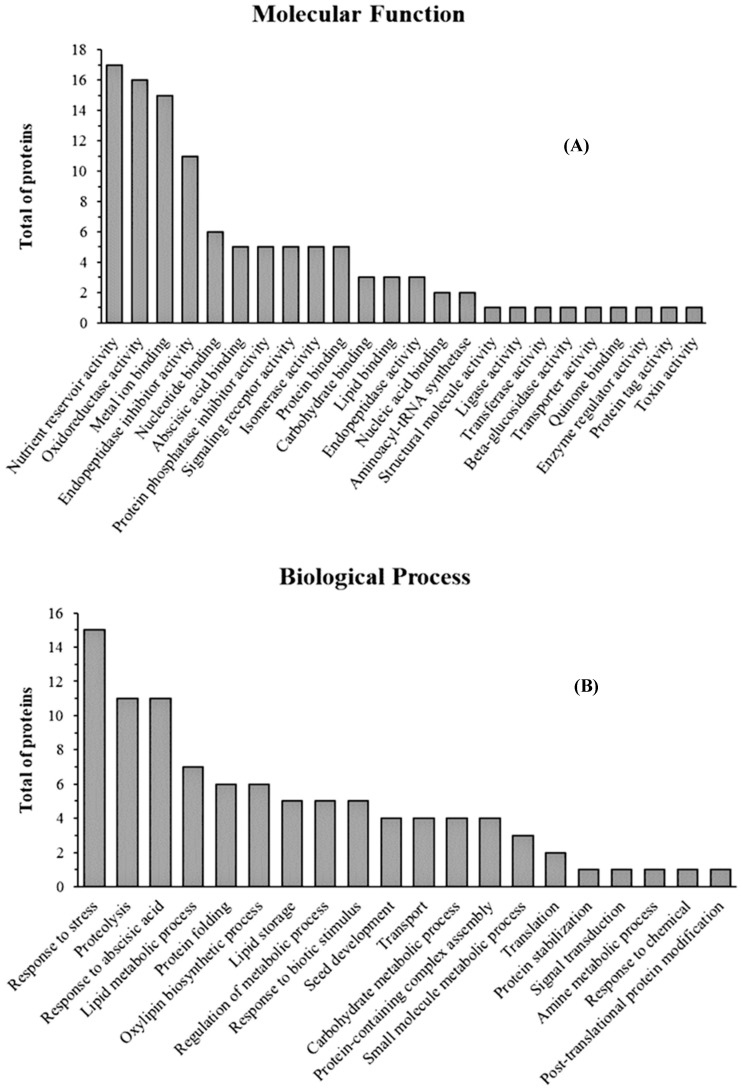
Molecular functions and biological processes of the proteins identified in the lunasin-enriched soybean extract (LSE). Molecular functions (**A**) and biological processes (**B**) were obtained from the UniProtKB and Gene Ontology databases, respectively. All identified information is presented.

**Figure 3 nutrients-16-02056-f003:**
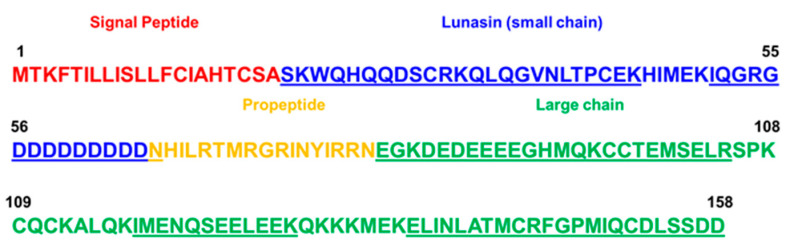
2S albumin sequence coverage by LC–MS/MS analysis. Each part of the sequence is identified and highlighted in corresponding colors. The sequence parts identified by MS analysis are underlined.

**Figure 4 nutrients-16-02056-f004:**
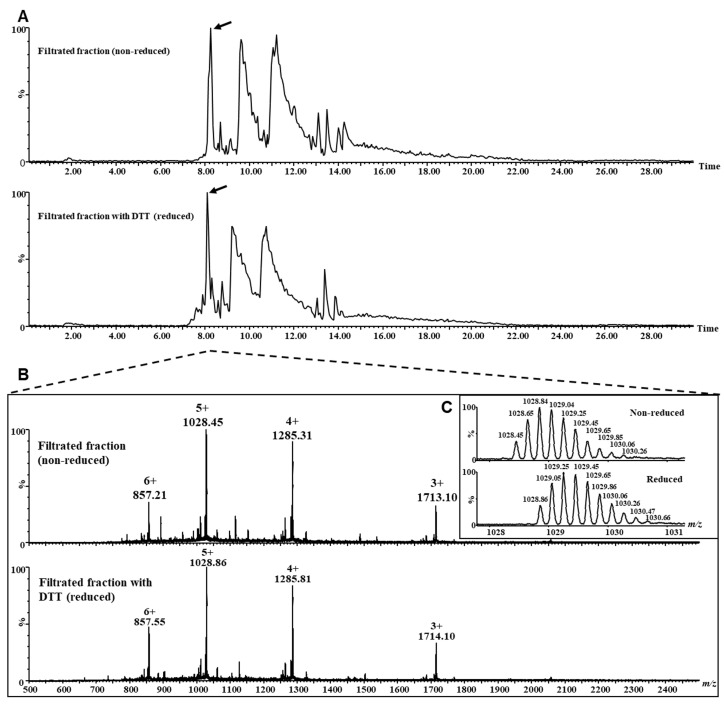
Analysis of the lunasin in the filtrated non-reduced and reduced lunasin-enriched soybean extract (LSE) samples by LC–MS. (**A**) Base peak ion chromatograms of the filtrated non-reduced and reduced LSE samples. Arrows indicate the time of elution of lunasin. (**B**) Mass spectra of lunasin in both non-reduced and reduced LSE samples from chromatograms at the times 8.26 and 8.11 min, respectively. (**C**) Comparison of the mass spectra of lunasin in both the non-reduced and reduced LSE samples for 5+ charged species.

**Figure 5 nutrients-16-02056-f005:**
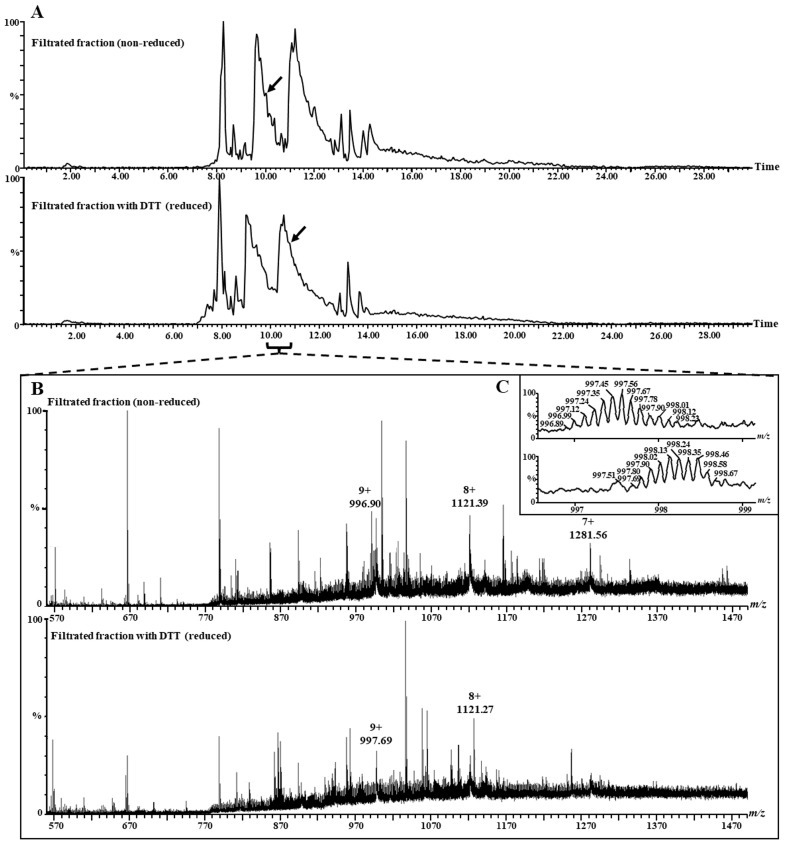
Analysis of the 2S albumin large chain in the filtrated non-reduced and reduced lunasin-enriched soybean extract (LSE) samples by LC–MS. (**A**) Base peak ion chromatograms of the filtrated non-reduced and reduced LSE samples. Arrows indicate the time of elution of the 2S albumin large chain. (**B**) Mass spectra of 2S albumin’s large chain in both the non-reduced and reduced LSE samples obtained from chromatograms at the times 10.05 and 10.99 min, respectively. (**C**) Comparison of the mass spectra of 2S albumin’s large chain in both the non-reduced and reduced LSE samples for 9+ charged species.

**Figure 6 nutrients-16-02056-f006:**
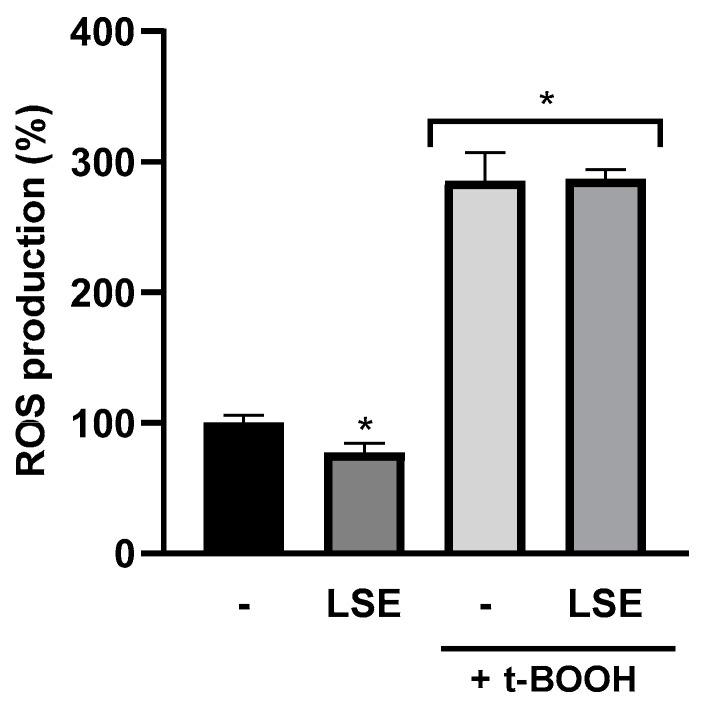
Antioxidant effects of lunasin-enriched soybean extract (LSE) in AGS cells. Effects of LSE (0.065 mg protein/mL) on reactive oxygen species (ROS) production (expressed as %) in AGS cells under basal conditions and chemically challenged with tert-butyl hydroperoxide (t-BOOH, 2.5 mM). The negative control indicates 100% ROS production. * *p* ˂ 0.0001.

**Figure 7 nutrients-16-02056-f007:**
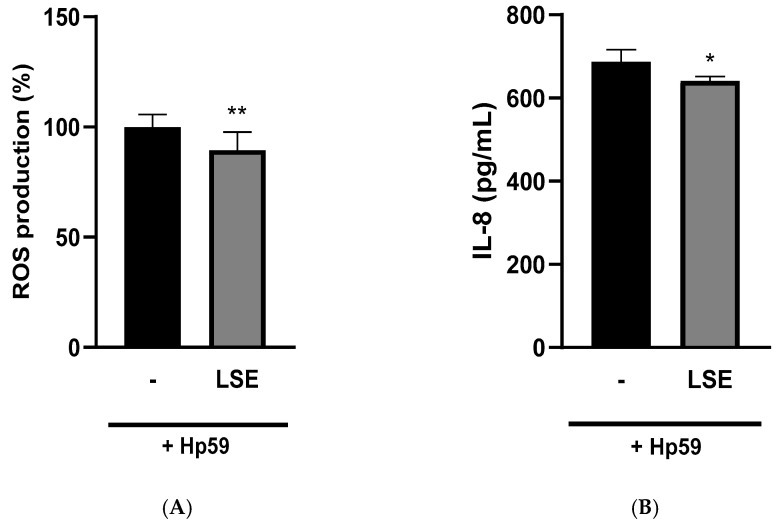
Effects of lunasin-enriched soybean extract (LSE, 0.065 mg protein/mL) on the (**A**) reactive oxygen species (ROS) production (expressed as % of ROS in comparison with non-treated infected cells, considered as 100%) and (**B**) interleukin (IL)-8 (expressed as pg/mL) in *Helicobacter pylori*-infected AGS cells. * *p* = 0.0222; ** *p* = 0.0087.

**Figure 8 nutrients-16-02056-f008:**
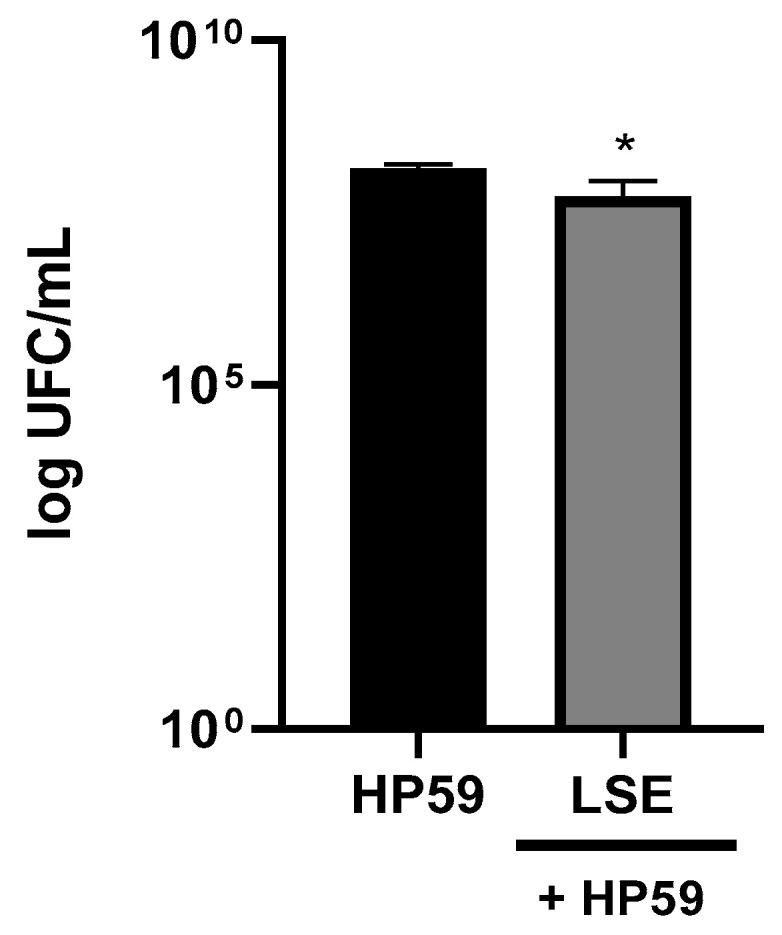
Antimicrobial effect of the lunasin-enriched soybean extract (LSE, 1 mg protein/mL) against *Helicobacter pylori*. The results are expressed as log CFUmL (mean ± SD) (n = 4). * *p* = 0.0034.

**Table 1 nutrients-16-02056-t001:** Quantification of proteins in the lunasin-enriched soybean extract (LSE) by mass spectrometry (MS) using the MS^E^ method.

Accession	Protein	Peptide Count	Unique Peptides	Confidence Score	Theoretical Mass (Da)	Total Experimental Amount (fmol)	Percentage in Total Protein (%)
P04776	Glycinin G1	83	48	801.50	56,333.71	8290.88	12.39
P0DO15; I1NGH2; K7KGR6; K7N005; P0DO16	Beta-conglycinin alpha subunit 2	75	54	813.70	70,591.40	7155.67	10.69
P02858	Glycinin G4	80	66	782.68	64,253.61	5943.14	8.88
P19594	2S seed storage albumin protein	32	32	235.11	19,030.31	5939.51	8.88
P11827	Beta-conglycinin alpha’ subunit	95	77	692.06	72,513.18	3939.93	5.89
A0A0R0GMV1	Glycinin G1 (fragment)	48	10	567.38	55,600.23	3816.23	5.70
P01070; A0A0R0IWE9; I1KYW9; Q39898	Trypsin inhibitor A	49	36	293.94	24,290.46	3659.55	5.47
K7LEQ5	Dehydrin	36	29	339.19	26,687.07	3482.17	5.20
Q9ZNZ4	Napin-type 2S albumin 1	33	32	255.07	18,404.98	1858.82	2.78
P04405	Glycinin G2	67	23	612.42	54,961.11	1593.32	2.38
P25974; A0A0R0I6G3	Beta-conglycinin beta subunit 1	70	29	505.48	50,532.97	1215.24	1.82
I1NGG4	Late embryogenesis abundant protein D-34-like	29	24	252.52	26,172.16	1027.04	1.53
P11828	Glycinin G3	48	18	537.14	54,869.09	1026.31	1.53
F7J077	Beta-conglycinin beta subunit 2	38	2	497.00	50,498.95	1021.06	1.53
P22895; O64458	P34 probable thiol protease	6	5	52.93	43,136.04	837.89	1.25
P01064; I1MQD2	Bowman–Birk type proteinase inhibitor D-II	12	12	123.67	10,323.16	769.52	1.15
P29531	P24 oleosin isoform B	13	7	140.98	23,392.69	736.24	1.10
P04347	Glycinin G5	51	12	460.16	58,412.39	730.70	1.09
P05046	Lectin	8	8	62.18	30,928.02	702.78	1.05
I1L957	Late embryogenesis abundant protein LEA	42	28	375.05	48,795.62	651.76	0.97

Only the 20 most abundant proteins are presented in the table. Theoretical mass corresponds to the precursor proteins. Total protein amount corresponds to the sum of all detected ions.

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
