# Peer review of "Proteomic Characterization of a Lunasin-Enriched Soybean Extract Potentially Useful in the Treatment of Helicobacter pylori Infection"

_nutrients, 2024, doi:10.3390/nu16132056_

Round 1

Reviewer 1 Report

Comments and Suggestions for Authors

In this study, the authors aimed to investigate about the antioxidant, anti-inflammatory and antibacterial properties against H. pylori infection of lunasin-enriched soybean extract (LSE), also performing a characterization of LSE by mass spectrometry. The study is interesting and can be provide reference data for further studies. In my opinion, the minor revisions suggested below:

Line 52. Please specify how SE was prepared.

Line 318. I suggest the authors to combine table 1 and 2 although they represent a qualitative and quantitative analysis respectively; the information reported are redundant and does not help reading. Furthermore, I also suggest inserting other information such as the intensity of the mapped peptide and score.

Line 356 and 591. Please correct the layout throughout the text

Line 410. Figure 3. it would be appropriate to add the numerical position of the aa in the sequence (at the beginning and at the end of the line)

Line 638-640. Not only the antimicrobial effect but also the antioxidant and anti-inflammatory effects could be attributed to other compounds of SE. Please rephrase the concept.

Author Response

In this study, the authors aimed to investigate about the antioxidant, anti-inflammatory and antibacterial properties against H. pylori infection of lunasin-enriched soybean extract (LSE), also performing a characterization of LSE by mass spectrometry. The study is interesting and can be provide reference data for further studies. In my opinion, the minor revisions suggested below:

1. Line 52. Please specify how SE was prepared.

Answer: As specified in the section 2.1. Materials and Methods, line 86, SE is an albumin-enriched soybean extract that was provided by Reliv International Inc. and the specific information on its preparation is confidential.

2. Line 318. I suggest the authors to combine table 1 and 2 although they represent a qualitative and quantitative analysis respectively; the information reported are redundant and does not help reading. Furthermore, I also suggest inserting other information such as the intensity of the mapped peptide and score.

Answer: To address this question and avoid redundancy, we chose to exclude table 1, since all data on the identified proteins are presented in table S1, and we adjusted the text and table numbers accordingly. Additionally, we included a table S3, which contains the intensity of the peptides and their respective scores, as suggested.

3. Line 356 and 591. Please correct the layout throughout the text

Answer: The layout has been corrected in the new version of the manuscript.

4. Line 410. Figure 3. it would be appropriate to add the numerical position of the aa in the sequence (at the beginning and at the end of the line)

Answer: As suggested, the numerical position of the aa in the sequence (at the beginning and at the end of the line) was added in the figure 3.

5. Line 638-640. Not only the antimicrobial effect but also the antioxidant and anti-inflammatory effects could be attributed to other compounds of SE. Please rephrase the concept.

Answer: As suggested, the concept was rephrased to include all the bioactivities assayed. (Conclusions section)

Reviewer 2 Report

Comments and Suggestions for Authors

1.     The introduction jumps between different topics without clear transitions, which can confuse readers. For example, it abruptly shifts from discussing the global prevalence of H. pylori to the molecular research on its physiopathological changes.

2.     The statement "the dynamics between H. pylori and the gut microbiota can influence systemic health" is not followed by a reference, leaving the claim unsupported.

3.     The claim that "diet therapy is often neglected" lacks context and evidence. This is a broad statement that requires more specific evidence or examples to support it.

4.     The text repeats information unnecessarily. For instance, it mentions the adverse side effects of antibiotics and the need for alternative approaches in both the context of eradicative treatment and later in discussing dietary influences.

5.     The phrase "demonstrated human health benefits" is vague and should specify the particular benefits being discussed. Similarly, "different criteria regarding the compounds involved in the antibacterial response" is unclear and should be explained.

6.     The statement "no data on their effects against H. pylori are still available" conflicts with previous mentions of antibacterial activity against H. pylori in soybean extracts. It creates confusion about whether any studies exist on this topic.

7.     There are grammatical issues such as "multifunctionality has been associated to its" which should be "associated with its" and "suggest a potential this peptide in the control" which should be "suggest the potential of this peptide in the control".

8.     The review mentions that soybean-derived peptides have antimicrobial potential but fails to provide details on the specific mechanisms or studies supporting this claim.

9.     A good methods section should provide enough detail so that another researcher could replicate the study. If steps are omitted or described too vaguely, this could be a significant issue.

10.  The procedures should be described clearly and logically. If the sequence of steps is confusing or out of order, it can be difficult for readers to follow and for other researchers to replicate.

11.  The methods chosen should be appropriate for the research question and justified within the text. If the chosen methods are not the best fit for the study's aims, or if there is no explanation for why certain methods were used, this is problematic.

12.  All materials used in the study should be listed, including specific details like the manufacturer and model number for equipment, or detailed descriptions of any surveys or questionnaires used. Missing this information can impede replication.

13.  The methods section should include details about how data were analyzed. If statistical methods are not clearly described, or if inappropriate statistical tests are used, this can undermine the study's findings.

14.  If the study involves human or animal subjects, there should be a section on ethical considerations, including approval from an appropriate ethics committee. If this is missing, it raises ethical concerns.

15.  Terms and definitions should be consistent throughout the methods section. Inconsistencies can lead to confusion and misinterpretation of the methods.

16.  Any potential sources of bias should be acknowledged, and steps taken to minimize bias should be described. If this information is missing, the reliability of the results can be questioned.

17.  Details on how the sample size was determined and how participants or samples were selected should be included. If this information is missing, it can affect the study's validity.

18.  If the study involves experimental conditions, there should be control conditions described to allow for comparison. Missing control conditions can lead to issues with interpreting the results.

Comments on the Quality of English Language

Moderate editing of englush language needed

Author Response

1. The introduction jumps between different topics without clear transitions, which can confuse readers. For example, it abruptly shifts from discussing the global prevalence of H. pylori to the molecular research on its physiopathological changes.

Answer:  In the new version of the manuscript the introduction was fully revised, and the text reformulated accordingly for a better reading and comprehension.

2. The statement "the dynamics between H. pylori and the gut microbiota can influence systemic health" is not followed by a reference, leaving the claim unsupported.

Answer: A new reference has been added to support the mentioned claim (line 47). 

3. The claim that "diet therapy is often neglected" lacks context and evidence. This is a broad statement that requires more specific evidence or examples to support it.

Answer: The sentence has been slightly modified and a reference has been added (line 57).

4. The text repeats information unnecessarily. For instance, it mentions the adverse side effects of antibiotics and the need for alternative approaches in both the context of eradicative treatment and later in discussing dietary influences.

Answer: The new version of the manuscript has been carefully checked and the repeated information has been removed from the text.   

5. The phrase "demonstrated human health benefits" is vague and should specify the particular benefits being discussed. Similarly, "different criteria regarding the compounds involved in the antibacterial response" is unclear and should be explained.

Answer: The sentence has been rewritten for a better understanding (line 67-69).

6. The statement "no data on their effects against H. pylori are still available" conflicts with previous mentions of antibacterial activity against H. pylori in soybean extracts. It creates confusion about whether any studies exist on this topic.

Answer: The sentences have been rewritten for a better understanding (lines 67-72)   

7. There are grammatical issues such as "multifunctionality has been associated to its" which should be "associated with its" and "suggest a potential this peptide in the control" which should be "suggest the potential of this peptide in the control".

Answer: The grammatical errors have been corrected (Lines 76 and 79).

8. The review mentions that soybean-derived peptides have antimicrobial potential but fails to provide details on the specific mechanisms or studies supporting this claim.

Answer: The observation was taken in consideration and the statement was rephrased, with the addition of a new reference regarding the mechanisms (line 72-74). 

9. A good methods section should provide enough detail so that another researcher could replicate the study. If steps are omitted or described too vaguely, this could be a significant issue.

Answer: The methods section was fully revised to improve the detail of the methodology described.

10. The procedures should be described clearly and logically. If the sequence of steps is confusing or out of order, it can be difficult for readers to follow and for other researchers to replicate.

Answer: The methods section and procedures used were fully revised to improve with new details of the methodology described.

11. The methods chosen should be appropriate for the research question and justified within the text. If the chosen methods are not the best fit for the study's aims, or if there is no explanation for why certain methods were used, this is problematic.

Answer: The methods section was fully revised for improvements on the clarity of the information (Line 235 for the viability assay, lines 273-278 for the measurement of ROS, and lines 291-293 for cytokines release).

12. All materials used in the study should be listed, including specific details like the manufacturer and model number for equipment, or detailed descriptions of any surveys or questionnaires used. Missing this information can impede replication.

Answer: The methods section was fully revised for improvements, adding possible missing information.

13. The methods section should include details about how data were analyzed. If statistical methods are not clearly described, or if inappropriate statistical tests are used, this can undermine the study's findings.

Answer: The statistical methods are described on the section 2.7 dedicated to the statistical analysis employed. For the proteomics, since it requires different parameters, the methods are described in the subsection 2.3.3.

14. If the study involves human or animal subjects, there should be a section on ethical considerations, including approval from an appropriate ethics committee. If this is missing, it raises ethical concerns.

Answer: There are no humans or animal subjects in the study, therefore there is no ethical considerations. The sentence (lines 249-250) has been modified to avoid any confusion.

15. Terms and definitions should be consistent throughout the methods section. Inconsistencies can lead to confusion and misinterpretation of the methods.

Answer: The methods section was fully revised for improvements and to address any inconsistencies.

16. Any potential sources of bias should be acknowledged, and steps taken to minimize bias should be described. If this information is missing, the reliability of the results can be questioned.

Answer: The potential sources of bias were considered and the methods were described and adjusted according to it.

17. Details on how the sample size was determined and how participants or samples were selected should be included. If this information is missing, it can affect the study's validity.

Answer: As indicated in section 2.2., the sample used in our study was a lunasin-enriched soybean extract which concentration for the cell culture experiments was determined according to its cytotoxic effects. No human or animals were used.

18. If the study involves experimental conditions, there should be control conditions described to allow for comparison. Missing control conditions can lead to issues with interpreting the results.

Answer: The control conditions used are fully described for each assay.